# Molecular Pathways in Prolactinomas: Translational and Therapeutic Implications

**DOI:** 10.3390/ijms222011247

**Published:** 2021-10-18

**Authors:** Betina Biagetti, Rafael Simò

**Affiliations:** Diabetes and Metabolism Research Unit, Vall d’Hebron University Hospital and Vall d’Hebron Research Institute (VHIR), Universidad Autónoma de Barcelona, 08035 Barcelona, Spain

**Keywords:** prolactinoma, molecular, review, prolactin, receptor, dopamine, lactotroph tumour

## Abstract

Prolactinoma has the highest incidence rate among patients with functional pituitary tumours. Although mostly benign, there is a subgroup that can be aggressive. Some clinical, radiological and pathology features have been associated with a poor prognostic. Therefore, it can be considered as a group of heterogeneous tumours. The aim of this paper is to give an overview of the molecular pathways involved in the behaviour of prolactinoma in order to improve our approach and gain deeper insight into the better understanding of tumour development and its management. This is essential for identifying patients harbouring aggressive prolactinoma and to establish personalised therapeutics options.

## 1. Introduction

Prolactinomas comprise 40 to 60% of all pituitary adenomas and are mostly presented in women of childbearing age as indolent microadenomas (<10 mm in diameter) [1,2]. However, there are great differences in the clinical behaviour depending on gender, age and size. In addition, it should be noted that prolactinomas are the second-most frequent aggressive pituitary tumours after corticotroph tumours [3].

Prolactinoma aggressiveness is defined as a radiologically invasive tumour which has an unusually rapid rate of tumour growth despite optimal standard therapies [4]. Baseline characteristics such as plasma prolactin (PRL) levels, tumour size or radiological features such as invasion, are not prognostic factors of aggressiveness per se [5]. Some clinical features such as younger age at diagnosis (<20 years) [6], especially in males [7] or resistance to dopamine agonists (DA) have been associated with a poor prognostic [8,9]. However, the early identification of prolactinoma aggressiveness and resistance to DA remains controversial [10,11] and, therefore, represents a therapeutic challenge.

In addition, pathology proliferation markers (Ki-67 expression: ≥3%, mitotic count n > 2) taken alone are not reliable markers of malignancy [12,13]. In fact, only the rare pituitary tumours with distant metastases can be named “pituitary carcinomas”, indicating truly malignant behaviour [14,15].

Overall, these findings reveal that prolactinomas can be considered as a group of heterogeneous tumours. Therefore, better understanding of the molecular pathways involved in tumour development is essential for identifying patients harbouring aggressive prolactinoma and establishing personalised therapeutic options.

## 2. Pituitary Development and Lactotroph Lineage

### 2.1. Anatomy and Ontogeny of Pituitary

The pituitary gland has a dual embryonic origin (neuroectodermal and non-neuroectodermal) that confers a unique histological appearance. Both parts are made up of anatomically and functionally distinct lobes of the pituitary. The posterior lobe (neurohypophysis) consists of nervous tissue arising from the embryonic forebrain and represents an extension of the hypothalamus. The anterior lobe (adenohypophysis) [16], which is derived from an outpouching of the roof of the pharynx, called Rathke’s pouch, can be further divided into three parts

Pars anterior or distalis: this is the largest part and is responsible for hormone secretion.

Pars tuberalis: this is an upwards extension of the pars anterior and wraps the pituitary stalk, which is composed of unmyelinated axons from the hypothalamic nuclei. The hormones oxytocin and vasopressin accumulate in these axons [16].

Pars intermedia: this is a thin epithelial layer that separates the pars anterior from the posterior lobe. Pars intermedia contains the colloidal matrix and includes the remainder of the Rathke’s pouch cleft. It has some pituitary stem cells, and secretes melanocyte-stimulating hormones and endorphins [16].

### 2.2. Lactotroph Lineage

During pituitary development, the dorsal and ventral sides generate signalling mediators which regulate transcription factors that are essential in governing cell proliferation and differentiation [17,18]. Lactotrophs, also called mammotrophs or prolactin cells, comprise 20–50% of the cellular population of the anterior pituitary gland depending on sex and physiological status [19]. These type of cells are acidophils and stain red by hematoxylin and eosin. Ontogenetically, lactotrophs belongs to the Pit-1-dependent lineage in the caudomedial region of the pituitary gland, together with somatotrophs and thyrotrophs, and produce prolactin, a polypeptide hormone prolactin of 199 amino acids (24 kDa) (Figure 1). The best known function of PRL is the growth and preparedness of the mammary gland for lactation but its exact role in the male is poorly understood [20].

## 3. Prolactin Axis, Dopamine Receptor and Prolactin Receptor

### 3.1. Prolactin Axis

PRL secretion is mainly under inhibitory stimuli, via tuberoinfundibular dopamine (TIDA) neurons in the hypothalamus binding to Type 2 Dopamine Receptor (D2R) and GABA (gamma aminobutyric acid), the latter playing a minor role [21]. PRL also controls its own secretion through a short loop negative feedback, stimulating TIDA cells [22] and in the own lactotroph cell by an autocrine loop [23] (Figure 2). There are also prolactin-releasing factors (PRFs) such as thyrotropin-releasing hormone, vasoactive intestinal peptide, serotonin, histamine, oxytocin and oestrogens [24]. Nevertheless, the nature of the physiological PRF is unclear [24]. Nipple stimulation, light, olfaction, and stress, stimulate prolactin secretion. Other neurotransmitters and neuropeptides can also modulate PRL secretion (for example endothelin, TGF beta1, angiotensin, somatostatin, substance P, neurotensin, calcitonin, EGF, natriuretic atrial peptide, bombesin, cholecystokinin, acetylcholine, and vasopressin) [25,26,27]. Among medications that increase serum PRL, dopamine receptor antagonists such as neuroleptics (i.e., sulpiride, haloperidol, chlorpromazine, risperidone) and antiemetic drugs (i.e., metoclopramide, domperidone) can elevate serum PRL at the same range that is usually detected in prolactinoma [28]. Serotoninergic and antihistaminergic drugs are less potent than antidopaminergic ones [25,29].

As we mentioned above, prolactin is responsible for milk production and breast development, but also acts in many other less known functions such as metabolic functions [30], immune functions [31] and the regulation of adult stem/progenitor cells [32] (Figure 2).

Circulating prolactin is low in males and nonlactating nonpregnant females, and a hypo-prolactinemic syndrome has not been described. Its overproduction could lead to galactorrhea and has an inhibitory effect on the release of gonadotropin-releasing hormone that results in infertility, menstrual cycle disturbances in women, and decreased libido and spermatogenesis in men, as well as other less studied effects that will be described below.

### 3.2. D2 Receptor, Prolactin Secretion and Antiproliferative Activity

The dopamine receptors are members of the superfamily of G protein-coupled receptors. There are five types, which include D1 to D5. Each receptor has a different function [33]. The five different dopamine receptors can be grouped in two categories depending on the action to the G protein:(1)D1 and D5, also called D1-like receptors. By coupling to G stimulatory sites, adenylyl cyclase cAMP is activated, which in turn activates protein kinase A (PKA), thus enhancing transcription. These receptors are abundant in the striatum, nucleus accumbens, olfactory bulb, and substantia nigra and are essential in regulating the reward system, motor activity, memory, and learning.(2)D2, D3 and D4 (D2-like receptors). By coupling to G inhibitory sites, they inhibit adenylyl cyclase and activate K+ channels. They are expressed mainly in the striatum, as well as the external globus pallidus, the core of the nucleus accumbens, hippocampus, amygdala, and cerebral cortex.

The regulation of the lactotroph cell is manly by D2R [34] (Figure 3).

When dopamine binds to D2R, both PRL secretion and lactotroph proliferation are inhibited. Within seconds after binding, dopamine activates K+ channels which leads to membrane hyperpolarisation and the inactivation of voltage-gated calcium channels. Consequently, a reduction of intracellular free calcium occurs, thus resulting in the inhibition of PRL release from secretory granules (Figure 3). Within minutes to hours, dopamine suppresses adenylyl cyclase activity and lowers inositol phosphate metabolism, resulting in the suppression of PRL gene expression. Within days, dopamine inhibits lactotroph proliferation and decreases the size of lactotrophs [33]. All this orchestrated process explains in part why in general, but not invariably, prolactin secretion and tumour volume are in parallel in prolactinomas.

### 3.3. Prolactin Receptor, Intracellular Signalling and Autocrine Function

Prolactin action has different outcomes in the endocrine, autocrine and paracrine signalling. In addition, the target cell and the different pathways of activation/inactivation determine its final action.

Classically, PRL is known to activate the peripheral prolactin receptor (PRLR), promoting proliferation and inhibiting apoptosis linked to its main function of milk production and breast development. How autocrine/paracrine or endocrine PRL levels collaborate with oncogenes to foster tumourigenesis, e.g., in breast tissue, but also in other hormonally responsive cancers, is not well understood [35]. In this section, we will focus on PRL central action in the pituitary gland and the main recognised downstream pathways activated after prolactin binding to its receptor on lactotroph cells.

PRLR is expressed on TIDA cells (a short-loop feedback circuit) and also on lactotrophs of the pituitary gland where it may provide an autocrine loop to regulate lactotroph function [23]. The PRLR is a member of the cytokine receptor superfamily that signals via Janus kinase-2-signal transducer and activator of transcription-5 (JAK2-STAT5), phosphoinositide 3-kinase-Akt (PI3K-Akt-mTOR) or the MAPK pathways to mediate changes in transcription, differentiation and proliferation [36] (Figure 3).

The JAK2-STAT5 pathway is involved in processes such as immunity, cell division, cell death and tumour formation [37]. Ferraris J. et al. [38] showed that acute hyperprolactinemia induced in ovariectomised rats using PRL injection or dopamine antagonist treatment rapidly increased apoptosis and decreased the proliferation of lactotroph cells, in contrast to the classical proliferative or antiapoptotic actions exerted by PRL in most other tissues [39]. The same group recently investigated this antiproliferative effect of PRL in the pituitary and identified PRLR/JAK2/STAT5 pathway constitutive activation in lactotroph cells as a major link in producing prolactin antiproliferative effects [40]. Thus, the constitutive paracrine/autocrine activation of the PRLR/JAK2/STAT5 pathway in the lactotroph cell inhibits cell proliferation and induces apoptosis as opposed to the classical proliferative effects of PRL (i.e., hormonally responsive tissues).

The mTOR pathway regulates the cell cycle and its overactivity has been associated with several cancers [41], as well as with aggressive pituitary tumours [42,43]. Gorvin et al., determined the PRLR sequence in 46 prolactinomas, and found that a PRLR variant was associated with increased signalling in this pathway, which was reverted with everolimus, a mTOR inhibitor [44].

The MAPK pathway is one of the best-defined pathways in cancer biology. It promotes cellular overgrowth activating proliferative genes, and, at the same time, enables cells to overcome metabolic stress by inhibiting AMPK signalling, a key sensor of cell energetic status [45]. Long-term activation of the Ras/MAPK pathway was found to promote differentiation of the bihormonal somatolactotrope GH4 precursor cell into a prolactin-secreting cell (lactotroph cell phenotype) in both in vitro and in vivo [46].

## 4. Tumour Development, Lessons from Mice

### 4.1. D2R-Deficient Mice and Dopamine Transporter-Deficient Mice

As we mentioned above, dopamine not only inhibits PRL secretion but also proliferation and hypertrophy of the lactotroph cells. The evidence supporting the antimitotic activity of dopamine on lactotrophs comes from the induction of pituitary hyperplasia in D2R-deficient mice (which prevents dopamine action) [47,48] and the opposite phenotype, the pituitary hypoplasia that exhibits the dopamine transporter (DT)-deficient mice (which increases dopamine availability) (Figure 4).

The major characteristics of D2R-deficient mice are chronic hyperprolactinemia and lactotroph hyperplasia which lead to adenoma development in aged females only. D2R null mice of either sex had 3- to 4-fold higher basal PRL levels, but without repercussions in spermatogenesis, and with a little impact on fertility in females. Older D2R-null female mice developed significant hyperplasia (up to 50-fold higher) and large lactotroph adenomas. However, the pituitary in age-matched males was similar or only double in size [47,48]. These findings led to the hypothesis that the proliferative action of prolactin is conditioned by an oestrogen permissive action. In addition, local growth factors (normally under dopaminergic inhibition) and an imbalance between angiogenic/antiangiogenic factors could participate by promoting the availability of different growth factors and mitogens [48].

DT-null mice have increased dopaminergic tone, anterior pituitary hypoplasia, dwarfism (D2R is present in somatotroph cells), and an inability to lactate [49]. DT elimination dramatically reduces the numbers of lactotrophs and somatotrophs in the pituitary. However, DT-null mice present two unexpected factors: an unchanged basal serum PRL level, and an unresponsiveness to the dopamine receptor antagonist sulpiride [33]. These events could be the result of compensatory mechanisms acting to diminish the effects of enhanced dopamine tone, such as downregulation of the lactotroph D2R and an increased sensitivity to PRL secretagogues.

Although animal models are a huge source of knowledge, they are not always identical to humans. The presence of the mutated D2R gene in humans was examined by direct DNA sequencing in 79 pituitary tumours, mostly lactotroph and mixed GH/PRL-secreting tumours. No mutations were demonstrated, and all migration abnormalities detected were due to polymorphisms within the D2R gene [50]. More recently, a retrospective case-control study analysing the frequency of five DRD2 polymorphisms in 148 patients with prolactinoma and 349 healthy subjects, failed to demonstrate any difference in genotypes between case and control group. In addition, a correlation between DRD2 polymorphisms and cabergoline responsiveness was not found in the prolactinoma group [51]. In addition, prolonged dopamine deficits in humans caused by neuroleptics, pituitary stalk dysfunction or direct hypothalamic damage, did not induce prolactinomas.

Collectively, these observations argue against the presence of mutated D2R or the loss of dopamine inhibition as primary causes of lactotroph tumours in humans.

### 4.2. PRLR Deficient Mice

Plasma PRL levels in the receptor-null mice (PRLR^−/−^) are increased 30-fold in males and 100-fold in females, and this is accompanied by a somewhat enlarged pituitary gland [33]. The double mutant (D2R^−/−^, PRLR^−/−^) mice used to bypass the short loop feedback and investigate possible dopamine-independent effects of PRL on lactotroph function, exhibited prolactinomas that were significantly larger than those observed in D2R ^−/−^ mice [52] However, PRLR^−/−^ mice presented more profound hyperprolactinemia and larger tumours than age-matched D2R ^−/−^ mice [52]. The associated phenotype in PRLR^−/−^ mice was more severe in females [53]. Furthermore, to determine whether a direct pituitary effect of hyperprolactinemia on lactotroph growth does exist, Schuff et al. [52], assessed lactotroph proliferation in vitro by culturing pituitary cells from wild-type and D2R^−/−^ mice in the presence of recombinant PRL. PRL treatment markedly inhibited the proliferation of wild-type female and male D2R^−/−^ derived lactotrophs but had no effect on female D2R^−/−^ lactotrophs. These findings suggest a downregulation or desensitisation of PRLR in response to chronic hyperprolactinemia.

Taking into account all this data, there is a cross-talk between dopamine and prolactin which seems modulated by gender factors (i.e., oestrogens).

## 5. Gender Differences and Oestrogens in Prolactinomas

Oestrogen was the first-discovered inducer of pituitary tumour transforming gene (PTTG) through its nuclear αER (oestrogen receptor α). PTTG is a well-known proto-oncogene whose aberrant accumulation is known to cause cancer development, activating fibroblast growth factor 2 and VEGF production, thus further promoting invasiveness and angiogenesis. In human and rodent pituitary, αER plays a major role in the physiological regulation of the secretory and mitotic activity of lactotroph cells, whereas in somatotroph cells, both αER and β are of importance [54]. αER expression in prolactinomas correlates with tumour size and resistance to treatment and is different regarding gender [55,56]. These facts have been confirmed in a clinical study showing a clear association between the low nuclear expression of αER (measured by immunohistochemistry) in prolactinomas and several clinical variables such as gender (low expression in males), tumour size, proliferative activity, low rate of surgical cure, DA resistance, and tumour progression [56]. More recently, these findings have been confirmed by the same method [57], and with more sophisticated techniques, including transcriptomic, microarray and comparative genomic hybridisation analyses [58]. Thus, αER immunostaining could be helpful as a prognostic marker in prolactinomas. The lower expression level of αER in male tumours may explain the higher risk of more aggressive tumour behavior, recurrence, and resistance to treatment.

More recently, Xiao Z et al. [59] investigated the effects of αER and PRLR signalling cross-talk in the bromocriptine-resistant prolactinoma cell line. Surprisingly, they found increased levels of ERα and PRLR protein expression in bromocriptine-resistant prolactinomas. In addition, a reciprocal positive regulatory loop that could contribute to bromocriptine resistance was reported. Furthermore, it was shown that αER inhibition restored bromocriptine sensitivity. It seems, therefore, that the ER pathway could be helpful not only for characterisation, but also as a therapeutic target (Figure 5).

## 6. Somatostatin Receptors and Prolactinoma

The inmunohistochemical analysis of somatostatin receptors (SSTR) in prolactinomas demonstrated that SSTR5 was the most frequent, followed by SSTR2A and SSTR1 [60]. Since SSTR5 is more important in PRL release regulation [61], somatostatin analogues with improved selective binding affinity for SSTR5 subtype may be effective in the treatment of hyperprolactinemia. Recently, some case reports have shown tumour shrinkage and prolactin level normalisation in resistant prolactinomas treated with pasireotide long-acting release, a second-generation somatostatin receptor ligand which binds to multiple SST receptors, but with a particularly high affinity for SST5 receptor [62,63]. These promising results should be confirmed in specific clinical trials.

## 7. Genomics in Prolactinoma

Prolactinomas can appear as a result of germline mutation present in multiple endocrine neoplasia, familial isolated pituitary adenomas (FIPA) [64] or Carney complex [65]. However, somatic mutations, as occur in other pituitary tumours such as corticotrophinoma, have only occasionally been reported [66,67]. Nevertheless, Li et al. [68] have recently identified a hotspot somatic mutation in splicing factor 3 subunit B1 (SF3B1) in up to 19.8% of prolactinomas. These patients with mutant prolactinomas displayed higher PRL levels and a shorter time in tumour development compared to patients without the mutation. Moreover, they identified that the SF3B1 mutation caused aberrant splicing of oestrogen-related receptor gamma, thus leading to a stronger binding of pituitary-specific positive transcription factor 1, resulting in a greater transcriptional activation of PRL. More interesting, this mutation was more frequent in males than in females (24.34% vs. 10.67%), and demonstrated that SF3B1 mutation was significantly associated with poor prognosis. This result contributes not only to the understanding of gender differences in the natural history of prolactinoma but could also open up a new therapeutic strategy.

### HMGA1 Gene

High mobility group A proteins (HMGA) modulate transcription by altering the chromatin architecture binding to amino-terminal regions and thereby regulate the transcriptional activity of several genes [69]. The expression of HMGA genes is high in malignant cells in vitro and in vivo [69]. Transgenic mice overexpressing HMGA1 [70] and HMGA2 [71] develop mixed prolactinoma and growth hormone pituitary adenoma.

## 8. Clinical Features Predicting Prolactinoma Response to DA

The therapeutic approach to prolactinoma is a current hot topic. Although at present there is a general consensus that DAs are the first line in both micro- and macroprolactinomas, the following evidence argues against this general recommendation: (I) up to 20% of macroprolactinoma could be resistant to medical treatment. (II) Normoprolactinemia after dopamine agonist withdrawal is only reached in 20–35% of cases and, therefore, most patients will need lifelong medical treatment [72,73]. (III) A recent meta-analysis showed that long-term remission rates were significantly higher after surgery than after medical treatment [74]. (IV) Data on the cost-effectiveness analysis revealed that TSS was more cost-effective than medical therapy not only in macroprolactinomas, but also in microprolactinomas in young patients with a life expectancy greater than 10 years [75].

Several studies have explored the early clinical predictors of the response to DA. Some of them argue in favour of prolactin normalisation as the strongest predictor to guide clinical decisions [76,77,78]. However, macroprolactinomas with very high basal levels of PRL could have significant PRL reductions despite not reaching PRL normalisation by the third month after the start of treatment and be good long-term responders [79,80,81]. In this regard, the percentage of prolactin reduction could be a more valuable predictor to response to DAs than a predefined cut-off value [73]. We, and others [82], argue in favour of a direct measure of tumour shrinkage response in the third trimester of treatment as an early marker of tumour response. This strategy permits us to take a more individualised decision and not delay foreseeable successful trans-sphenoidal surgery (TSS) in a subgroup of poor responders to DAs with eventual long-term aggressive behaviour [81]. Taking into account that macroprolactinoma is the second-most frequent aggressive tumour [3] predominantly in young men, continued DA dose escalation and extended medical treatment should be carefully balanced with the possibility of performing an early and successful TSS. From the research point of view, TSS samples obtained by TSS have a unique value for investigating the underlying molecular pathways associated with clinical phenotypes, which is essential for gaining new insights into personalised treatment.

## 9. Future Direction and Medical Options in Aggressive Prolactinomas

Knowledge of the molecular pathway allows us to improve our understanding of the mechanisms of tumour growth/aggressiveness. Beyond DAs, currently, we have no approved drugs for treating prolactinoma. Therefore, all efforts to increase our knowledge of the underlying molecular mechanisms of prolactinoma aimed at designing more personalised therapeutic strategies are very welcome.

Table 1 summarises the future therapeutic options for aggressive prolactinoma based on the available evidence. The following pathways seem closely related to prolactinoma aggressiveness.

### 9.1. JAK2-STAT

As mentioned above, this is a major pathway in pituitary PRLR. Although PRL has extrapituitary proliferative actions, constitutive activation in lactotroph cells by the JAK-SATAT pathway acts as a proapoptotic and antiproliferative factor [40].

Atiprimod, an anticarcinogenic agent targeting STAT3, was effective in apoptotic induction in GH3 pituitary adenoma cells, a model of the lactotroph cell [83]. Therefore, this kind of drug could be useful in aggressive prolactinoma. However, clinical trials are required to confirm this hypothesis.

### 9.2. PI3K-Akt-mTOR

Aydin et al. [84] studied the miRNA-mediated drug repositioning (transcriptome data that exploit disease-specific signatures in addition to biological and pharmacological data to elucidate a rational prioritisation of pathways and drugs) in 17 prolactinomas. The group found seven drugs including 5-fluorocytosine, nortriptyline, neratinib, puromycin, taxifolin, vorinostat, and zileuton as potential candidates for the treatment of prolactinoma. Except for puromycin, the other six drugs act through the PI3K/Akt pathway. They also demonstrated the inhibition of proliferation with such drugs in the PRL-producing MMQ tumour cell line. These findings confirm that PI3K/Akt is an important pathway in prolactinoma development and show the therapeutic potential of drugs targeting this pathway.

Everolimus, an mTOR inhibitor, was able to revert increased mTOR signalling in certain variants of PRLR that have constitutively activated these pathways [44]. 

Although everolimus has been employed in several aggressive neuroendocrine tumours, its use in pituitary tumours is not standardised and has been limited to a few case reports [3,85,86].

### 9.3. MAPK/AMPK Pathway

As mentioned above MAPK/AMPK have an interlink related to cell proliferation and energetic status [45]. Recently, Ding et al. [87] using the model of DRD2−/− mice found that the blockade of MAPK14 expression in mice significantly reduced prolactinoma formation and PRL production and secretion. This highlights MAPK14 as a potential therapeutic target in the treatment of prolactinoma.

Likewise, in the past 10 years, metformin (an antidiabetic drug) has been attracting increasing interest due to its anticancer effects [88]. These effects are exerted by stimulating AMPK. Indeed, some studies performed in human lactotroph cell cultures showed that metformin reduced lactotroph cell proliferation and promoted their apoptosis [89]. However, a recent prospective study performed in 10 adults with cabergoline-resistant prolactinoma, in which metformin (1.0–2.5 g/d) was added to cabergoline, failed to show a consistent inhibitory effect in serum prolactin levels; unfortunately, tumour volume changes were not reported [90].

### 9.4. Oestrogen Modulation

As previously indicated, prolactinomas have oestrogen receptors which induce the formation of pituitary adenomas in sensitive rats or mice [91]. In humans, we have some evidence regarding the oestrogen influence. In this regard, prolactinomas are more frequent in young women. Likewise, in the Dutch transgender registry, there was found a higher risk of prolactinomas in transwomen, compared to the general Dutch female population [92]. However, this result was not confirmed by other groups, despite prolonged oestrogen exposure [93].

On the other hand, the lower expression of the αER level in male tumours in comparison with female tumours seems to confer a higher risk of more aggressive tumours, recurrences, and resistance to treatment in males [56,57,58]. In addition, Choudhary et al. reported that treatment with raloxifene, (an oestrogen-receptor modulator) was associated with an up to 25% decrease in the PRL level in 10/14 (71%) patients with prolactinoma who were on stable doses of DAs, while two patients (14%) normalised their serum prolactin levels [94]. The mechanism by which a low expression in αER in males confers a greater risk of a poor response to DAs and of recurrences is not fully understood.

**Table 1 ijms-22-11247-t001:** Future therapeutic options for aggressive prolactinoma based on the available evidence.

	Place of Action	Evidence (References)	Clinical Trials
Capecitabine and Temozolomide in firstline	MGMT inhibits DNA synthesis and slows growth of tumour tissue	Isolated human case reports summarised in [86]	Ongoing NCT03930771 for functional and non-functional aggressive pituitary tumours
Pasireotide	multireceptor ligand SSTR5 > SSTR2 > SSTR3 > SSTR1	Case reports (humans) [62,63]	No
Atiprimod	JAK2-STAT → STAT3	rat cell lines GH3 [83]	No
5-fluorocytosine, nortriptyline, neratinib, taxifolin, vorinostat, zileuton	PI3K-Akt-mTOR	MMQ cell lines and mRNA-miRNA data integration [84]	No
Everolimus		prolactinoma derived cells (human) [44] Case reports (humans) [85]	No
Blockade of MAPK14	MAPK/AMPK	mice and human prolactinoma cells [87]	No
Metformin		prolactinoma derived cells (human) [8]	No. A pilot study (*n* = 10) failed to show PRL normalisation (no data on tumour growth)
Raloxifene	oestrogen receptor modulator	case reports (humans) [94]	Pilot study (*n* = 14), not randomised, no control group
Immunotherapy	PD-L1 PIT-1	case reports (humans) [95]	No
	Ipilimumab and nivolumab		Progressive pituitary adenoma/carcinoma NCT04042753 and NCT02834013

### 9.5. Temozolomide and Others Cytotoxic Agents

Temozolomide (TMZ) is used as a first-line chemotherapeutic agent for aggressive pituitary tumours and carcinomas [4]. TMZ exerts its cytotoxic activity by alkylating DNA at the O6-methylguanine DNA methyltransferase (MGMT) position of guanine resulting in irreversible DNA damage and cell death. A lower expression of MGMT counteracts the effects of TMZ, and its expression correlates with the effectiveness of the drug [4]. For this reason, the routine determination of MGMT status in all aggressive pituitary tumours by immunochemistry is recommended [4]. To the best of our knowledge, apart from TMZ there is no other chemotherapeutic agent for treating aggressive prolactinoma in the first line, except for few cases of the synergic combination of capecitabine and TMZ (CAPTEN) either in TZM naïve patients or after TZM fails and confined case reports, of TMZ association with VEGF-targeted therapy (bevacizumab or apatinib) [3,86].

Turchini et al. [95] have recently found that the programmed death-ligand 1 (PD-L1) expression was common in somatotrophs, lactotrophs, and PIT-1 positive plurihormonal pituitary adenoma. These results open up a new potential role of immunotherapy as an adjuvant treatment of selected cases of prolactinoma which needs to be explored. In this respect, two clinical trials with ipilimumab and nivolumab are actively recruiting patients with aggressive pituitary tumours/carcinoma NCT04042753 and NCT02834013 (Table 1).

## 10. Concluding Remarks

Current guidelines in prolactinoma management [96] strongly recommended DAs (preferably cabergoline) as the first line of treatment. This is valid for micro- and macroprolactinomas. Surgery is reserved for when there is resistance to high doses of cabergoline or it is not well tolerated. However, biomarkers of response to DAs such as tumour shrinkage at the third month of treatment seem to better predict the long-term response, thus allowing for more personalised treatment to be implemented.

The guidelines for aggressive tumours [4] recommend surgery as first line treatment and the adjuvant use of radiotherapy in patients with relevant tumour growth despite surgery with pathological markers (the Ki67 index, mitotic count, p53 immunodetection). The unique formal recommendation as first-line chemotherapy after surgery is temozolomide in monotherapy with MGMT status evaluation to predict the response. This shows that we are far from achieving a personalised approach to prolactinoma.

An early TSS in the subgroup of patients with potential aggressive tumours allows us to investigate the underlying molecular pathways associated with clinical phenotypes. In this regard, some PRLR variants that increase PRL secretion and lactotroph proliferation could benefit from the mTOR inhibitors such as everolimus. The assessment of the SSTR, PD-L-1 and MGMT status in tumour tissue will provide the mechanistic basis for recommending more targeted therapies, resulting in more personalised and cost-effective treatments.

There is an urgent need for basic and clinical researchers to join forces to gain more insight into the underlying molecular mechanisms of prolactinomas. This approach will permit us to improve clinical practice and provide a better approach to the treatment of prolactinomas, in particular, the aggressive ones.

## 11. Search Strategy and Selection Criteria

We searched PubMed for articles published, with the terms “prolactinoma [tiab] AND molecular [tiab]”, “prolactinoma [tiab] AND aggressive [tiab]”, “prolactin receptor [TI] AND review [Filter]” and “dopamine receptor” [TI] AND review [Filter], “aggressive pituitary tumours [tiab]”. Articles identified by these searches and relevant references cited in those articles were reviewed. Only articles published in English were included. Review articles and book chapters are also cited to provide readers with more detail in some specific areas not addressed in this review. We largely selected those published in the past 20 years but did not exclude seminal older articles.

## Figures and Tables

**Figure 1 ijms-22-11247-f001:**
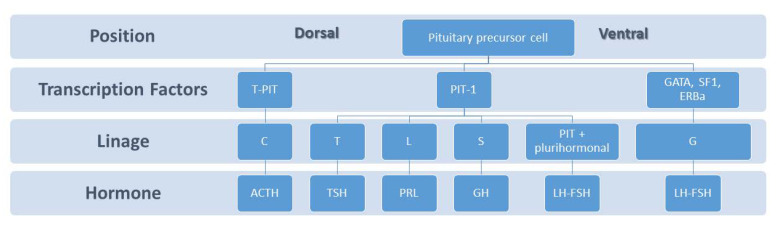
The dorsal and ventral side of the embryonic pituitary generate proliferative and positional signals which regulate the expression of transcription factors. The T-PIT signal differentiates the most dorsal cells into corticotropes (C). *Pit1* is induced in the caudomedial region of the pituitary gland, which ultimately gives rise to somatotropes (S), lactotropes (L), and thyrotropes (T). On the ventral side when GATA, SF1, ERBa are activated, the gonadotrope lineage (G) is determined.

**Figure 2 ijms-22-11247-f002:**
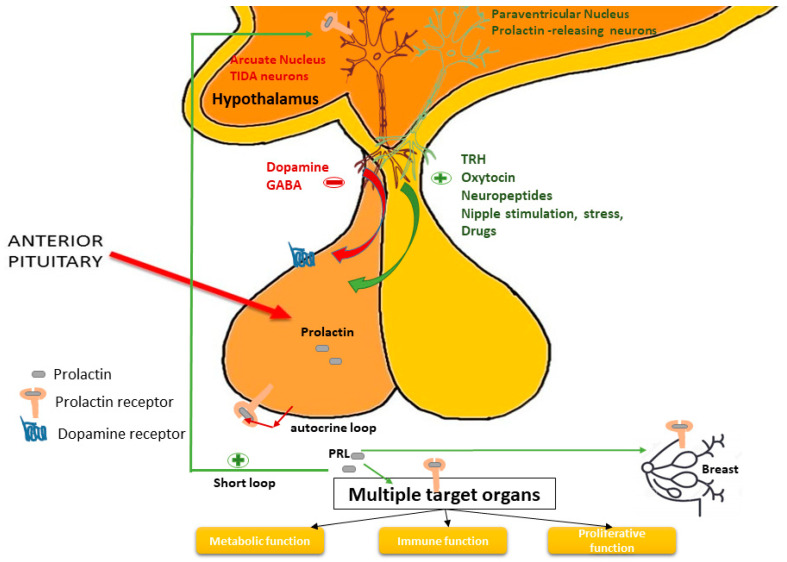
Control of prolactin production and secretion. Prolactin controls its own secretion through a short loop, stimulating TIDA cells and in the own lactotroph cell by an autocrine loop.

**Figure 3 ijms-22-11247-f003:**
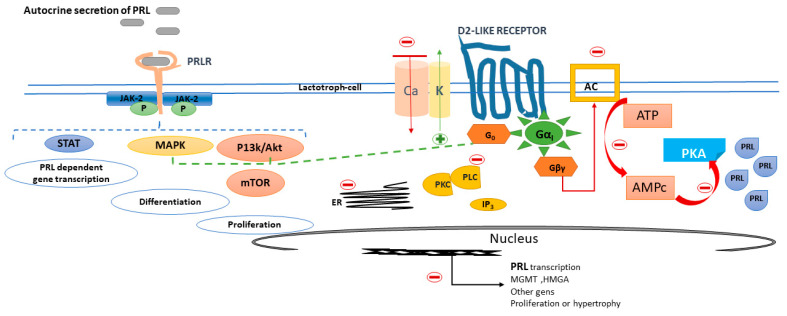
Prolactin and dopamine receptor in the lactotroph cell. After dopamine binding to dopamine receptor type 2, K+ channels are activated and adenylyl cyclase activity is inactivated, resulting in the suppression of PRL gene expression, lactotroph proliferation and a decrease in the size of hypertrophied lactotrophs. D2 via G0 also activates phosphatidylinositol 3-kinase (PI3K), and mitogen-activated protein kinase (MAPK) pathways to prevent lactotroph proliferation. The autocrine released prolactin binds to the prolactin receptor and, via the Janus kinase-2-signal transducer and activator of transcription-5 (JAK2-STAT5), (PI3K-Akt-mTOR) or the MAPK pathways, mediates changes in transcription, differentiation and proliferation.

**Figure 4 ijms-22-11247-f004:**
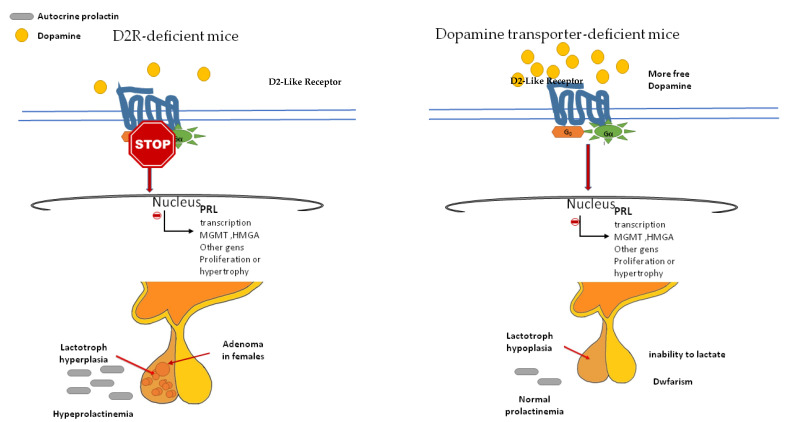
Dopamine receptor-deficient mice and dopamine transporter-deficient mice. D2R-deficient mice, led to hyperprolactinemia, lactotroph hyperplasia in male and adenomas in female. Dopamine transporter-deficient mice have an increased dopaminergic tone due to dopamine availability. D2R: dopamine receptor.

**Figure 5 ijms-22-11247-f005:**
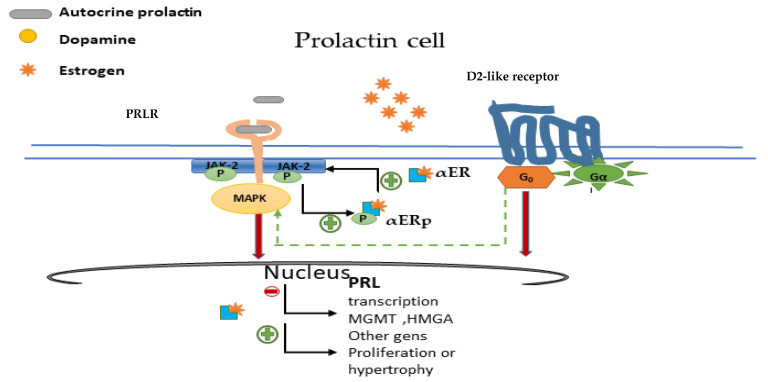
Nuclear oestrogen receptor (αER) and crosstalk with PRLR and D2R. Oestrogens bind to αER, which stimulates lactotroph cell growth and proliferation. Prolactin induces phosphorylation of αER (αERp), while oestrogen promotes PRLR upregulation via pERα. ERα inhibition, restores pituitary adenoma cell sensitivity to bromocriptine by activating among others D2-G0 MAPK signalling. PRLR: prolactin receptor; D2R: dopamine receptor; αERp: phosphorylated αER.

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
