# Peer review of "Molecular Pathways in Prolactinomas: Translational and Therapeutic Implications"

_ijms, 2021, doi:10.3390/ijms222011247_

Round 1

Reviewer 1 Report

The authors perform an exhaustive revision of the molecular mechanisms of the pathogenesis of lactotroph tumors, intending to link them with future therapeutic alternatives. Despite the difficulty of summarizing such big information, the authors have constructed an ordinate paper. 

Only minor suggestions:

The terminology of pituitary tumors changes several times in the text, e.g., Prolactinoma, prolactinome. I suggest all the terminology referent to pituitary tumors using the term Lactotroph tumors, gonadotroph tumors... and the same with the different lineages: corticotroph lineage, gonadotroph lineage...

There are a number of errors in the wording of the text which should be carefully modified. I attach the document with suggestions

Page 6, lines 209-210: Collectively, these observations... Change the wording of this sentence.

Author Response

Reviewer #1:

Comments and Suggestions for Authors

The authors perform an exhaustive revision of the molecular mechanisms of the pathogenesis of lactotroph tumors, intending to link them with future therapeutic alternatives. Despite the difficulty of summarizing such big information, the authors have constructed an ordinate paper.

Answer: Many thanks for the revision process and your kind comments on our paper.

Only minor suggestions:

The terminology of pituitary tumors changes several times in the text, e.g., Prolactinoma, prolactinome. I suggest all the terminology referent to pituitary tumors using the term Lactotroph tumors, gonadotroph tumors... and the same with the different lineages: corticotroph lineage, gonadotroph lineage...

Answer: We agree with the reviewer and we have homogenized the terminology and ‘lactotroph tumors’  has been the predominant used term . However, taking into account that both basic researchers and clinical endocrinologists will be the main readers of this review, we would prefer to maintain the term “prolactinoma” in some parts of the text because it is currently used in clinical practice.

There are a number of errors in the wording of the text which should be carefully modified. I attach the document with suggestions.

Answer: Thank you very much for taking your time to improve our manuscript. We followed all your recommendations. In addition, the manuscript has been revised again by a native English speaker.

Page 6, lines 209-210: Collectively, these observations... Change the wording of this sentence.

Answer: This sentence has been replaced by the following: “Collectively, these observations argue against the presence of mutated D2R or the loss of dopamine inhibition as primary causes of lactotroph tumors in humans”. Page 6 lines 227-228

Reviewer 2 Report

The authors in the present paper aimed to resume state of art on molecular pathways in prolactinomas. The aim is well defined and interesting; however, the current form has some major issues that make it unacceptable for publication in IJMS.

-the manuscript has evident English mistakes and need to be extensively written and corrected

-mat and meth section should be added containing research criteria used for the selection of publications considered for the present review (keywords, time considered, type of paper, searching databases included for the search)

-the paper is repetitive in some parts and has overlapping information between paragraphs and figures (i. e. lines 97-100 and figure 2, but it is not the only example)

-the paper lacks significant citations that support the topic explained in some specific paragraphs (lines 32-35, 58-61, 75-77, 77-81, 264-265, and others). Following this, the citations should be, if possible, about original articles and not referring to other less recent reviews.

-the aim is to clarify the molecular pathways involved in prolactinoma. I think that the author should be more precise and detailed about that. Some examples in the 3.2 paragraph, where no info has been given to Dopamine receptor type 3, even if mentioned with the other 4 Dopamine receptors. For paragraphs 4.1 and 5, a complementary figure visualizing and resuming the pathway would help

-I think that the 7.1 paragraph is out of context and does not add any significant information about genomics in prolactinoma. If you think it should be useful, please better explain this topic

Author Response

Reviewer #2:

The authors in the present paper aimed to resume state of art on molecular pathways in prolactinomas. The aim is well defined and interesting; however, the current form has some major issues that make it unacceptable for publication in IJMS.

Answer: Many thanks for the revision process and the criticism of our paper. We have addressed point by point all your recommendations.

-the manuscript has evident English mistakes and need to be extensively written and corrected

Answer:  The revised manuscript has been extensively revised by a native English speaker.

-mat and meth section should be added containing research criteria used for the selection of publications considered for the present review (keywords, time considered, type of paper, searching databases included for the search)

Answer:  Thank you very much for pointing out this important omission. A new section: ”Search strategy and selection criteria” has been added to the revised manuscript. Page 13 lines 480-488  

-the paper is repetitive in some parts and has overlapping information between paragraphs and figures (i. e. lines 97-100 and figure 2, but it is not the only example)

Answer: The referee is right in indicating that some parts of the manuscript are repetitive. Consequently, we have eliminated overlapping information given in Figure 2  page, Figure 3 page 4  and deleting section 7.1 pages 8-9 lines 316-322

-the paper lacks significant citations that support the topic explained in some specific paragraphs (lines 32-35, 58-61, 75-77, 77-81, 264-265, and others). Following this, the citations should be, if possible, about original articles and not referring to other less recent reviews.

Answer: Following the reviewer’s recommendation, we have added the following seminal citations of original articles to the revised manuscript:

12 Delgrange, E.; Sassolas, G.; Perrin, G.; Jan, M.; Trouillas, J. Clinical and Histological Correlations in Prolactinomas, with Special Reference to Bromocriptine Resistance. Acta Neurochir (Wien) 2005, 147, 751–757; discussion 757-758, doi:10.1007/s00701-005-0498-2

15 Watanabe, Y.G. Effects of Brain and Mesenchyme upon the Cytogenesis of Rat Adenohypophysis in Vitro. I. Differentiation of Adrenocorticotropes. Cell Tissue Res 1982, 227, 257–266, doi:10.1007/BF00210884.

16 Chen, R.P.; Ingraham, H.A.; Treacy, M.N.; Albert, V.R.; Wilson, L.; Rosenfeld, M.G. Autoregulation of Pit-1 Gene Expression Mediated by Two Cis-Active Promoter Elements. Nature 1990, 346, 583–586, doi:10.1038/346583a0.

18 Brown, R.S.E.; Kokay, I.C.; Phillipps, H.R.; Yip, S.H.; Gustafson, P.; Wyatt, A.; Larsen, C.M.; Knowles, P.; Ladyman, S.R.; LeTissier, P.; et al. Conditional Deletion of the Prolactin Receptor Reveals Functional Subpopulations of Dopamine Neurons in the Arcuate Nucleus of the Hypothalamus. J Neurosci 2016, 36, 9173–9185, doi:10.1523/JNEUROSCI.1471-16.2016

19 Jimenez, A.E.; Voogt, J.L.; Carr, L.A. Plasma Luteinizing Hormone and Prolactin Levels and Hypothalamic Catecholamine Synthesis in Steroid-Treated Ovariectomized Rats. Neuroendocrinology 1977, 23, 341–351, doi:10.1159/000122683.

27 Sellers, Z.P.; Bujko, K.; Schneider, G.; Kucia, M.; Ratajczak, M.Z. Novel Evidence That Pituitary Sex Hormones Regulate Migration, Adhesion, and Proliferation of Embryonic Stem Cells and Teratocarcinoma Cells. Oncol Rep 2018, 39, 851–859, doi:10.3892/or.2017.6108.

These references have been highlighted in yellow in the revised manuscript (marked file).

-the aim is to clarify the molecular pathways involved in prolactinoma. I think that the author should be more precise and detailed about that. Some examples in the 3.2 paragraph, where no info has been given to Dopamine receptor type 3, even if mentioned with the other 4 Dopamine receptors. For paragraphs 4.1 and 5, a complementary figure visualizing and resuming the pathway would help

Answer: Thank you for your comments. Following your recommendation, we have added to the revised manuscript a more comprehensive description of Dopamine receptors, localization and action (page 4   lines 120-127).  We have also complemented paragraphs 4.1 and 5 with two new figures (Fig. 4 and Fig. 5 of the revised manuscript).

-I think that the 7.1 paragraph is out of context and does not add any significant information about genomics in prolactinoma. If you think it should be useful, please better explain this topic

Answer: We agree that this paragraph does not add any significant information and,  consequently, it has been eliminated.

Reviewer 3 Report

This is a nice and comprehensive review on etiopathogenesis of prolactinomas with an overview of treatment strategies, challenging present guidelines and offering insight into new therapheutic options. 

However, I would suggest implementing a Table into your manuscript summarizing new therapheutic options for agressive prolactinomas and available evidence. 

Please, revise your key words, there are only three, and they are incomplete.

Also, English language needs substatial editing, there are numerous spelling and grammar errors throughout the manuscript and in its present form it is not acceptable for publication. 

Author Response

 Reviewer #3:

This is a nice and comprehensive review on etiopathogenesis of prolactinomas with an overview of treatment strategies, challenging present guidelines and offering insight into new therapheutic options. 

Answer: Many thanks for the revision process and your kind comments on our paper.

However, I would suggest implementing a Table into your manuscript summarizing new therapheutic options for agressive prolactinomas and available evidence. 

Answer: Thanks!! This is a great idea! We have added a Table summarizing the future therapeutic options and available evidence (Table 1 of the revised manuscript).  

Please, revise your key words, there are only three, and they are incomplete.

Answer: Thanks for your comment; we have added 3 more key words: prolactin; receptor, dopamine and lactotroph tumours.

Also, English language needs substatial editing, there are numerous spelling and grammar errors throughout the manuscript and in its present form it is not acceptable for publication. 

Answer: The manuscript has been sent for English editing and proofreading

Round 2

Reviewer 2 Report

Second revision

I would thank the authors for the extensive editing of the previous version.

Hovewer the present revised version of the article is by myself still unsuitable for publication in IJMS. Please see the attached pdf with your latest version of  the manuscript in which my newest observations have been added. The article will be ready for publication only if these corrections are provided in the final editing

Author Response

 Many thanks for the second revision process and for giving us the opportunity of further improve our paper with your constructive criticism. As recommended, we have addressed all the indicated points and have included appropriate comments according to your suggestions.  The new text has been highlighted in yellow and the new references in pink. 

Reviewer 3 Report

The revised manuscript is now acceptable for publication. 

Author Response

We really appreciate your review and evaluation

Round 3

Reviewer 2 Report

the manuscript is now acceptable (there is only a minor mistake: progressive should be changed in PROGRESSIVE ( PAG 13, TABLE1, IN iMMUNOTHERAPY SECTION)